# Seven Steps to Strategic SDG Sensemaking for Cities

**Ville Taajamaa [1,\*], Minna Joensuu [1,2]** 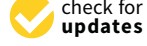**, Barbara Karanian [3] and Luis Bettencourt [4]**

1   Mayor's Office/Strategy, City of Espoo, 02720 Espoo, Finland; minna.joensuu@espoo.fi
2   Department of Health and Social Management, University of Eastern Finland, PL 1627, 70211 Kuopio, Finland
3   Stanford University, Stanford, CA 94305, USA; karanian@stanford.edu
4   Mansueto Institute for Urban Innovation, Department of Sociology and Evolution and Department of Sociology, University of Chicago, Chicago, IL 60637, USA; bettencourt@uchicago.edu
\*   Correspondence: ville.taajamaa@espoo.fi; Tel.: +358-40-508-3806

**Abstract:** This practitioner paper is based on the need to make sense of UN Agenda 2030 and Sustainable Development Goals (SDGs) at the city level and in an urban context. We examine the need to explain how to utilise the SDGs in strategic, tactical and operative urban development. We find that there are knowledge and practise gaps in how to localise SDGs in the urban context. This need and the lack of existing tools has led to the development of a strategic sensemaking process, which has been tested and developed with municipal and other practitioners, locally and globally. The paper presents findings from this process of development and from implementation pilots, including an SDG Sensemaking Tool (SST), a step by step iterative procedure to address these gaps. The main focus of this paper is the SDG Sensemaking process, which relies on analysing SDGs in relation to any given phenomena or project within or outside a city. The first results in this work-in-progress show that it contributes to an understanding on the complexity of how SDGs are related to the analysed phenomena, and catalyses the SDG localisation process, which helps make sense of how to navigate and measure progress in such complex environments. More research and applications are, however, needed, so as to further understand how urban governance can meet holistic, sustainable-development needs. Future work will, firstly, comprise further integrating SDGs into city-level strategies with a focus on the local, regional, national, and global impact on sustainable development and the actualisation of SDGs, and secondly, on further developing SST so that it can serve these purposes.

**Keywords:** 2030 UN Agenda & SDGs; localizing SDGs; urban sustainable development; strategic sensemaking; storytelling and strategy

## 1. Introduction

What makes a city sustainable and why is this important? These are fundamental questions for which there are now well-established answers based on systemic knowledge. Environmental, human-centred and natural sciences all answer these questions from their own general perspectives (Bettencourt 2021; Haarstad 2016; Hassan and Lee 2015; IPCC 2021). How to achieve sustainable, smart, inclusive, resilient, and prosperous cities in practice is, however, a context-driven question so that answers must be more than a sum of other approaches. For example, any successful strategy must include an affective and emotional buy-in from stakeholders, as well as cognitive rational actions. Any local practical approach is also replete with complex synergies and trade-offs between different goals and other dimensions of development (see, e.g., Lai 2020; Moyer and Bohl 2019; Pradhan et al. 2017). Due to its all-encompassing, systemic and complex nature, specifying the right questions and measurable quantities on sustainable development is at least as important as finding answers (Bettencourt 2013; Hölscher 2019). In other words, to navigate complex situations we must first understand the nature of the complexity underlying a given phenomenon, which can only be done in part via science and technology.

In practice, we rely on approaches that work in one setting and encourage us to simplify based on linear reasoning all too often, but this tends to lead to poor results (Snowden and Boone 2007). The alternative calls for grounding and resolving the complexity of processes and goals via discussions and collective learning between actual stakeholders. Once the complexity of the given phenomenon is understood in context, a sensemaking process on how to proceed towards solutions can start. This rationale, applied to urban development, provided the impetus to develop the SDG Sensemaking Tool (SST). It is described here as a practical process that can create an accurate and holistic view of a particular phenomenon in relation to all dimensions of sustainability, including economic, ecological, social, and cultural. In addition, SST is a facilitated process, where a set of indicators is developed to link specific local contexts to more general objectives. Then, these local context-driven metrics can be communicated and compared with internationally agreed sustainable development goals (SDGs), and with their specific targets and indicators. In addition to these two roles, SST has several other functions, described in more detail in Section 4.

Globally, cities are key stakeholders in creating sustainable development and combating climate change, as well as promoting human well-being and social equity. They are drivers of improvements in vitality, economic growth, and employment, and centres of governance, commerce, innovation, and social interaction (UNECE 2020; UN Habitat 2016, 2020). Thus, cities have fundamental roles to play in global sustainable development and, specifically, achieving the SDGs.

It is estimated that two-thirds of SDG indicators have urban components (OECD 2021). Many of the sustainability challenges we now face are the product of urbanisation, and so the solutions must be (Bettencourt 2015). More urban environments are also estimated to suffer greater impacts from challenges such as climate change and the loss of biodiversity than other areas (IPCC 2021). While these challenges need to be urgently resolved, new challenges such as global pandemics, refugee crises, political instability, more extreme weather, and other ecological, economic and social stresses are arising at an accelerating pace (IPCC 2021). Thus, cities need to quickly create new processes whereby ecological, economic, social, and cultural sustainability can be better combined with local living environments that stimulate equitable social interaction, embrace innovation, and serve as engines for the local and global economy (see, e.g., Harrison et al. 2010; Alberti et al. 2019).

Achieving sustainability is as much, if not more, about motivation, intention and participation than purely technical solutions. The goals are important but the journey must be feasible and inspiring. SST is a process of analysis that has tangible strategic, tactical and operative context-driven outcomes, which can be utilised to further the achievement of SDGs in cities, and contribute to overcoming some of these challenges. This article describes the advent of the SST in Espoo, Finland, including its structure and objectives along with general motivations and use cases.

The article is structured as follows. After presenting our research design and the journey of SST creation, we present a brief overview of the theoretical framework and background to explain relevant features of urban governance. Although the main goal of the article is to share experiences of a work-in-progress practitioner-based model and process, the theoretical background helps to frame the context and challenges for which the model is designed. We then shed light on what actions should be taken to better manage and lead sustainable development. Section 4 describes the sensemaking process task by task and presents the preliminary findings from the pilots. In Section 5, the Conclusions answer the research questions, present the state of the tool, and discuss future plans and development directions.

## 2. Research Design and Questions

### 2.1. Process and Story behind the SDG Sensemaking Tool

The epistemology of this practitioner article is founded on a pragmatic worldview (Brydon-Miller et al. 2003), and the point of departure for our research is the problem

of localizing the Sustainable Development Goals (SDGs) to an urban context. First, we selected research and analysis tools applicable to this specific problem, and decided to start the process by creating a draft version of the SDG Sensemaking Tool (SST). Next, we started a systematic testing process using expert sessions and workshops. The SST was modified according to the experiences and feedback generated in these workshops (Please see Figure 1).

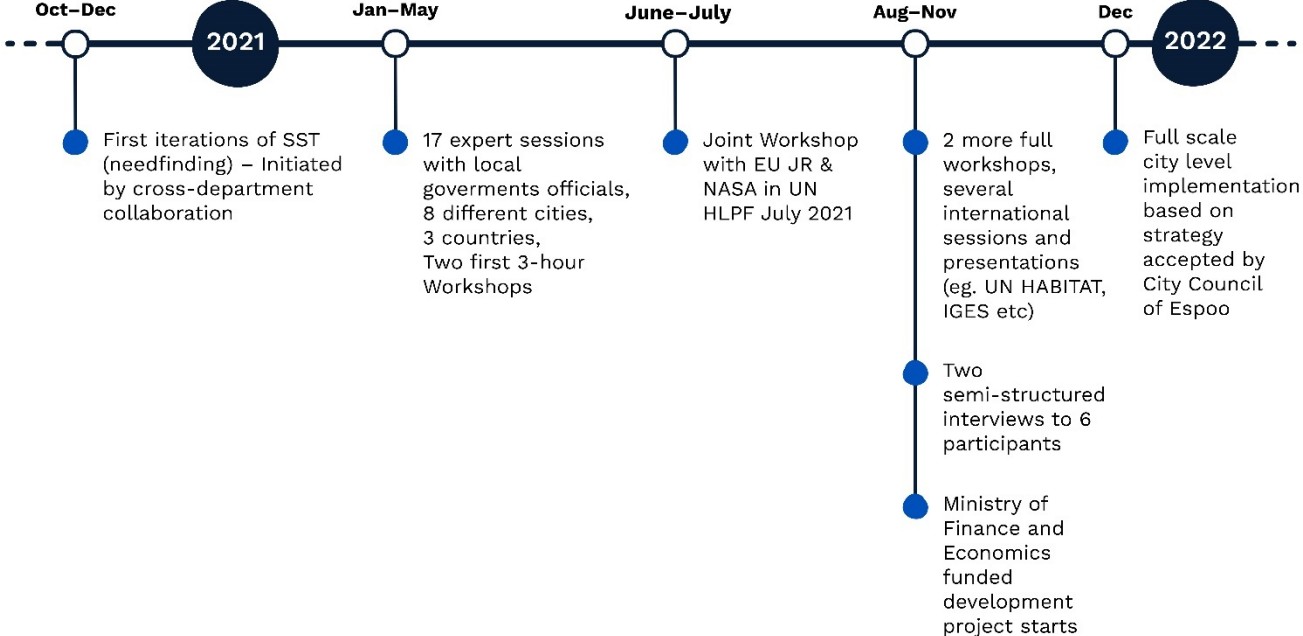

**Figure 1.** Timeline of SST development process.

In our context, the SDG Sensemaking Tool (SST) was developed in the City of Espoo[1], Finland, to provide the city administration, citizens and key partners with a holistic yet simple process connecting the everyday life of city officials and their work to SDGs and city strategy, "the Espoo story" (Fatima et al. 2021; Halko 2021). The main practical challenge was how to integrate SDGs into a city's strategy and daily operations, so that citizens, administrators and other partners would adopt and commit to it.

To this end, Espoo conducted a Voluntary Local Review (VLR), built on recognizing phenomena in a participatory and strategy-based manner (Espoo 2020; Tsolov and Geisselmann 2021). In the course of the review, the practical aim was to promote SDG integration at the city level, including systems-level operations such as procurement, communication, and budgeting. However, we noticed that in the process of cross-departmental collaboration, there were no pragmatic, concrete tools to specify to different parts of the city organization, citizens, and partners what the SDGs meant at a local level. This need was well aligned with the global situation within the urban context, and related to the ongoing discussion of 'Localizing the SDGs'. In an attempt to systematize our process, this led us to produce the first prototypes of the SST during the late autumn and winter 2020.

Tool development started with the first prototype during the autumn and winter of 2020 (Figure 1). Altogether, 17 different expert sessions and two full workshops were held during the spring of 2021. During the following summer, the SST was introduced in the UN HLPF (High-Level Political Forum) meeting, in the same workshop with the EU JRC (European Union Joint Research Centre) and NASA (National Aeronautics and Space Administration). Work continued into autumn 2021, with more workshops and interviews, including several international sessions, being held. As a result of this cumulative process, the SST is now being developed by a consortium of the six largest cities in Finland.

Experts from different domains of sustainable development within the city organisation provided feedback and comments on the early versions of SST at the beginning of 2021,

and the first pilot workshops were held on 24 January 2021. Altogether, 17 expert sessions were held between 12 January and 1 March to test the SST. These sessions lasted from 15 min (quick feedback reviews and comments) to 3 hour workshops. In all the sessions except one, the participants were city officials. After each session, the comments, criticism and experiences were discussed and evaluated, and the draft tool was updated accordingly.

In March 2021, the process was ready for international feedback and demonstrations. This consultations with experts in sustainable development and UN Agenda 2030, were held together with the inclusion of the cities of Utrecht in the Netherlands and Bristol in the UK. It was at this point that our SST was named 'Seven Steps to SDG Sensemaking'. By August 2021, there had been four full workshops lasting around three hours each, and numerous feedback sessions with experts from Europe, the USA, and East Asia. Approximately six months after the first workshops, semi-structured interviews were conducted with six different participants from two different workshops. Insights from these sessions are presented in the Learnings (Section 4.3).

There are many manuals and guidebooks on how SDGs should be reviewed and why it is important (EU 2021). Reviewing SDGs is an important part of creating strategic, tactical and operative understandings on what should be developed and why. Voluntary Local Reviews (VLRs), for example, are an acknowledged format for these processes, though their usability is case-dependent. The ways to further sustainable development in each city with its context specific needs and boundaries vary. This was another motivation for creating the SDG sensemaking tool: City officials and policy makers can utilise the SST analysis process in a context-driven way. This makes it possible to better understand and describe in detail what can and should be done locally to achieve the SDGs.

### 2.2. Research Questions

Strategic decision-making processes are essential to realizing sustainable development. However, achieving results in everyday work in cities requires the thorough implementation of sustainable development strategies and roadmaps (Kettunen et al. 2019; Fox and Macleod 2021). Localizing SDGs into strategic, tactical and operative activities in a participatory way is a vehicle for realizing all these objectives, but raises several strategic-level questions that the SDG Sensemaking process seeks to address. Thus, the SST has been created as an instrument to explain what the SDGs mean to a city as a whole. It is meant to create a clear link between a city's strategy, its strategic, tactical and operative goals and the 17 SDGs, and their targets and indicators over time. In this way, political and administrative leadership can make better sense of, measure, steer, and communicate strategic city-level goals using SDGs across scales, locally, regionally, nationally, and internationally.

Therefore, we posed an overarching research question, which stems from both local and global sustainable development needs: "How to utilise UN Agenda 2030 and SDGs, which are a multilateral, global, national, and state level set of goals, within an urban context, that is, in a city?" More precise questions were formulated as the results analysis progressed. We realized that it is important to analyse three, more precise, aspects of the process:

(1) What does it mean for a city to be committed to SDGs and their measurement?
(2) How can a city use SDGs to make sense of, measure, monitor, and communicate both its present and future plans through SDGs aimed at nation states and the global level?
(3) How does the tool manage to first make sense of and then reduce the process complexity, and thereby make the wicked problems of (sustainable) development simpler and more actionable?

### 3. Dilemmas of Planning Sustainable Development

#### 3.1. Cities, Complexity and Wicked Problems

Large-scale urbanization began in the 18th century alongside industrialization. Today, more than half of the world's population lives in cities, and it is estimated that by 2050 this share will rise to over 70 percent. Simultaneously, the global population is projected

to grow over this period by more than two billion people (United Nations Population Division 2020), especially in Asia and Africa. The ongoing urbanization trend highlights the importance of cities in concentrating and solving global problems. They play a major role in creating and utilizing integrated societal changes and combating associated adverse effects (Bettencourt 2021).

Cities face complexities on at least three levels. First, cities themselves are complex networks of people, organisations and infrastructure, developing in open-ended ways (Bettencourt 2021). In the 1960s, cities were recognized as "complex systems whose infrastructural, economic and social components are strongly interrelated and therefore difficult to understand in isolation" (Jacobs 1961). From a researcher's perspective:

> " . . . there are almost as many perspectives on the nature of a city as there are persons
> researching their structure, managing their organizations, or engaging with their design".
> (Batty 2018)

First, research and scholarship since then have taught us that cities are organic, multiscalar systems with their own identities and capacities to respond to the demands and needs of their inhabitants. They are knowable systems with specific properties, of interacting people and social organizations in densely built spaces, serviced by infrastructure and managed by social and political organizations (Bettencourt 2013). As cities grow, they become increasingly complex, with more exacting tradeoffs between costs and benefits, new technologies, and new organizational forms.

Second, the urban sphere of activity is also complex in many respects. Cities and their decision makers operate in an environment that is challenged by global, national, regional, and local or organisational influence and reform needs. These challenges are often layered, oblique and contradictory (Niiranen and Joensuu 2014). This kind of complexity stems from, among other things, the interdependencies of a wide field of activity, the interests of people, and different ways of understanding the (urban) world and the state of change (Vartiainen et al. 2013).

Third, many of the challenges that cities face today and those that will be faced in the future are complex and systemic. Correspondingly, many of the issues that need to be addressed, for example within the scope of sustainable development, are "wicked". Wicked problems are phenomena that are difficult to solve because it is hard to verify the origin of the difficulties, and because of circular forms of causality in the form of vicious and virtuous cycles of change (Bettencourt 2014). For this reason, attempts to solve such problems tend to produce new challenges, and are never final (Vartiainen et al. 2013; Puustinen and Jalonen 2020). Wicked problems result in part from the fact that people have a limited ability to predict the future and understand the relationships between entities and their components (Rittel and Webber 1973). Systems that internalize and manage problems in a timely fashion are necessary to address this kind of challenge.

### 3.2. UN Agenda 2030 and SDGs as a Framework, Metrics and a Vocabulary

UN Agenda 2030 and the Sustainable Development Goals (SDGs) are not an exact or all-encompassing approach to global ecological, economic or social sustainable development challenges. The UN Agenda is an international political agreement that puts forward a list of objectives and a plan of action for the benefit of people, peace, the planet, and prosperity, achieved in partnership between all stakeholders. SDGs are interconnected and form a holistic set of goals (17 in total), targets (169) and indicators (230) that specify and assess the UN Agenda's objectives. SDGs are both qualitative and quantitative in nature, and represent a global perspective on how to ensure a good life for all people in a way that is sustainable for the planet. SDGs have all levels of organization and agency (strategic, tactical and operative targets and indicators) embedded (Pradhan et al. 2017; Moyer and Bohl 2019).

The interconnected nature of SDGs means that achieving one goal might affect the achievement of others. The same applies to different dimensions of sustainability. They can potentially be contradictory, involve trade-offs, pose challenges to each other, or demon-

strate synergies as they can also reinforce one another. A pragmatic example of tradeoffs is how to sustain or create economic growth for all with limited planetary resources. SDGs cannot be decoupled or focused on just one dimension. They function as a global framework and, while cities are increasingly global players, SDGs also have a national, regional, and most importantly local role in achieving sustainable development. Although SDG targets and indicators have explicit urban components (Goal 11), the need for more context-driven targets and indicators including all other goals is widely acknowledged (Costa et al. 2021; Immler and Sakkers 2021; Lai 2020; Pradhan et al. 2017; Moyer and Bohl 2019).

That said, UN Agenda 2030 and the SDGs constitute an excellent multilateral framework to understand and specify global and holistic sustainable development, and their measurement, management and communication. Compared to many other standards or ways of measuring accountability, SDGs are widely known, not only to government officials but also companies, NGOs, and even citizens. Although originally developed for nation states, SDGs are a good fit for cities too, offering local governments at the frontline of sustainable urban development a blueprint for action.

*3.3. Towards Sustainable Urban Governance*

Sustainable development is high on the agenda of many cities for good reasons (Gandini et al. 2021). Cities have a high impact on our everyday quality of life and range of choices such as how we move, where we work, and what kind of environment we live in. This multifaceted, networked complexity has implications for urban governance. Constructive interaction between the city and its inhabitants enhances the quality of life within and outside city borders (Harrison et al. 2010). How a city is planned, managed and governed has, for example, important implications for how it will cope with the impacts of climate change (Gandini et al. 2021). Complexity and wicked problems are a challenge for cities, but can be navigated by approaches developed in urban environments.

Tackling complex problems requires a set of approaches based on appreciation of the massively interconnected character of urban social and infrastructure networks, and their dynamics over time (Bettencourt 2015; Bettencourt 2021). A simple top-down approach is not enough. Attempts to better manage cities in only one or some of the spheres of complexity often fail. However, understanding the world as an organic system where change is continuous across scales can help to resolve this issue (Raisio and Vartiainen 2020). Cities will need to evolve in response to their inhabitants' changing needs and aspirations, and respond to the ideals envisaged in the global sustainability agendas (Alberti et al. 2019).

To help cities respond to sustainable development challenges, one of the key requirements for change relates to the mindset and attitudes of city officials and political leaders. A survey of Finnish municipal officials in 2017 asked the respondents to prioritize different aspects of sustainability. Economic sustainability was considered the most important, with 85.8% of respondents ranking it first. Social sustainability (71%) ranked second, ecological sustainability (49.5%) third, and the lesser known cultural sustainability factor polled fourth (Kettunen et al. 2019). The results reflect how, in Finland, the organization of city services has traditionally focused on the economic imperative, and the other three aspects of sustainability easily command less attention. A more balanced view of the four aspects would probably be beneficial to achieving the SDGs, and is explicit in their formulation.

The importance of sustainable development is becoming manifest in its increasing visibility in city strategies and other documents. To varying degrees, cities gather information, make long-term plans, and cooperate with other cities. However, civil society actors outside the city organization have played a lesser role (Kettunen et al. 2019). Effective strategies are based on a cooperation with stakeholders (Jabareen 2013; Evans et al. 2006), and cross-sectoral facilitation within the city organization is also important to implement strategies and plans. Central questions are how to make decision-makers, businesses and citizens genuinely enthusiastic about SDGs, and how to embed sustainable development in

their everyday work. The full scope of sustainable development is difficult to determine, and involves a wide range of uncertainties and value conflicts.

As mentioned above, UN Agenda 2030 was drawn up for implementation by nation states, so there must be a localisation process for it to function as a quantitative and qualitative set of indicators at the local level. Localising SDGs, including targets and indicators, requires a context-sensitive approach, and hence the ability to link global goals, targets and indicators to local objectives. This is the main reason why we created our SST. Next, we examine the SST process step by step.

## 4. Seven Steps to Strategic SDG Sensemaking

### 4.1. SDG Sensemaking 101

UN Agenda 2030 and the SDGs can be used as a framework to understand sustainable development challenges across the planet, and measure how nation states and other institutions such as cities are progressing towards their associated targets and indicators. Agenda can also be used as a vocabulary to communicate essential goals, rights, and objectives agreed by all nations in the UN system. The overall aim of the SST process is to explain how someone's work in each local context relates to and advances the SDGs (see Figure 2). Selected phenomena were analysed via SST, and he process results were translated into new, localised SDG indicators that are clearly linked to both local goals and global targets and indicators.

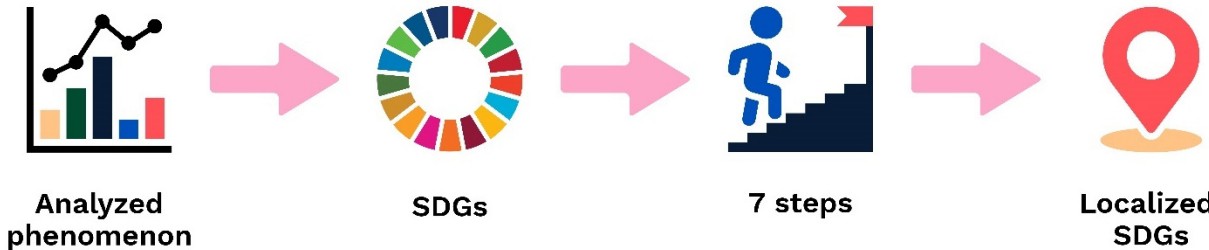

**Figure 2.** Abstract-level description of the SST process. Analysed phenomenon goes through a sensemaking process where it is reflected in relation to SDGs through a sensemaking process. In this section, we present the SDG Sensemaking tool, explaining each explicit step separately. The estimated time taken per step is based on learnings from pilot workshops, and represents a median of experiments. Thus, it is more indicative than exact or binding.

The tool can be applied to individual, team, unit, or department level phenomena or projects. They can be descriptive, such as 'building a new bike lane to location A', or thematic, for example 'developing a sustainable and smart urban area in location B'. The SST process can be executed individually or in teams. The latter is advisable, since a team possesses more diverse abilities and knowledge. The meta-level goal of the whole process is to create focused interactions amongst participants and across different stakeholders.

### 4.2. SDG Sensemaking Process Step by Step

This chapter describes the sensemaking process step by step to allow for deeper understanding of the process and the tasks included in it.

Step 1 "Defining your work in relation to the SDGs"

The first step comprises one task. The SST process begins by identifying a phenomenon, which must (a) connect with local and identified objectives, and (b) be understood within the SDG framework. The phenomenon or project selected for the analysis process must have strategic relevance and be either in the implementation phase, or in the presence of a concrete project or action plan. It should also have relevance in the selected time frame (e.g., year 2025 in the City of Espoo). Then, one SDG that best describes the phenomenon and two to four supporting SDGs are selected.

Time allocated for the task: approximately 20 min.

Step 2a "Different dimensions of sustainability"

The second step comprises three tasks (a, b, c). The goal is, first, to rank the dimensions of sustainability in relation to the phenomenon, and second, to explore the complexity of sustainable development in relation to the operative environment. The first task (2a) ranks the dimensions of sustainability (economic, ecological, social, cultural) according to their importance in relation to the phenomenon chosen in Step 1. This requires selective and convergent thinking, as in many cases the phenomenon is holistic and comprises all dimensions of sustainability.

Time allocated for the task: approximately 5–10 min.

Step 2b "Dimensions mapped through SDGs"

In task 2b, the SDGs selected in step 1 are inserted into sustainability dimensions. Each SDG can be inserted into only one dimension, but a single dimension may host several SDGs. Additionally, this task requires selection that is convergent and to some extent runs against the holistic nature of many SDGs. The idea is to think thoroughly and make the (sometimes difficult) choices to simplify the complex phenomenon, so as to make challenges more comprehensible and proposals easier to implement.

Time allocated for the task: approximately 10 min.

Step 2c "Main operational environment"

Task 2c is to choose the main level for the operational environment. 'Local' means 'city level' and can be a city organisation, partner or citizen, but actualised at the local level. 'Regional' means the region around a city, a description or definition that varies from country to country. National and international levels are linked to nation state operations and global bodies such as the UN, European Union, or other international institutions. In this task, the chosen phenomenon is analysed in relation to the levels of the operational environment. The nature of the phenomenon may vary significantly across the different levels. For example, an upper secondary course can be significant at the local level as a part of a curriculum, regionally as a matter of collaboration, nationally as an experiment for the development of future teaching, and internationally as a new concept. This can be elaborated on in a matrix form (Please see Figure 3).

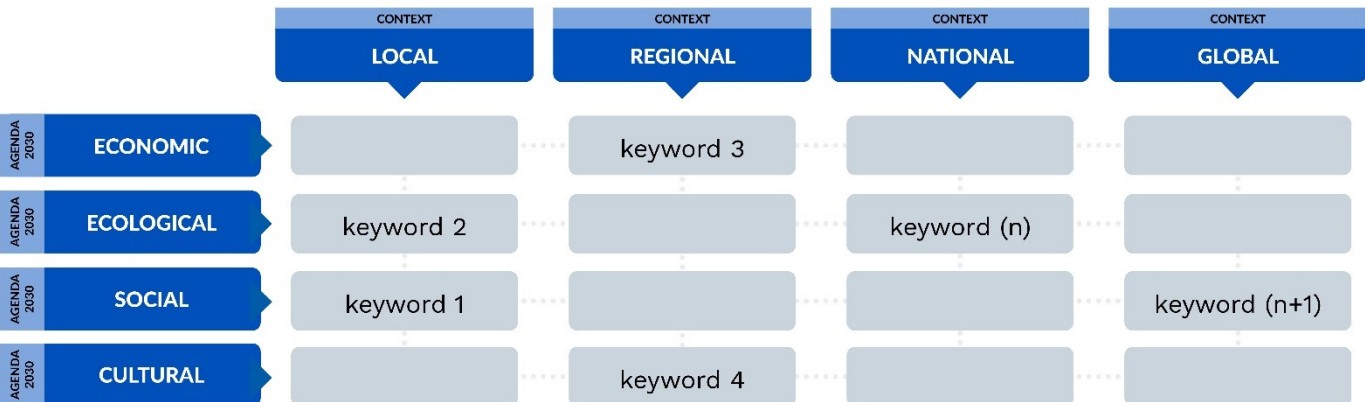

**Figure 3.** In Step 3, the researched phenomenon is investigated in relation to different dimensions of sustainability, and to their context.

Time allocated for the task: approximately 5–10 min.

Step 3a "Analysing dimensions and context together"

The third step is divided into two tasks (a and b). Steps 1 and 2 select the main and supporting SDGs, reveal the holistic nature of sustainable development, and explore the phenomenon's operational environment. In task 3a, the results of steps 1 and 2 are combined

in a matrix (Figure 3). The analysis is deepened by adding key words describing the phenomenon in the matrix. That is, the ranked dimensions are specified with a qualitative element. The key words can include, for example, the main projects, goals, or actions taken. They should be very concrete and descriptive to deepen understandings and explicate the decisions made in previous steps.

This is one of the key tasks of the whole process, after which it is advisable to have a break in the workshop program, typically from 70 to 90 min.

Time allocated for the task: approximately 30–45 min.

### Step 3b "Looping back to the start"

Task 3b provides an opportunity to reflect on how the SDG Sensemaking Process has progressed thus far, and, if needed, to rethink the main and supporting SDGs. If the chosen SDGs are changed at this point, it does not undermine the next steps of the process. It is wise to complete the ongoing process using the SDGs chosen in Step 1, and insert any new iterations for steps 1–3 after completing Step 7. This encourages learning.

Time allocated for the task: approximately 1–10 min.

### Step 4a "Role of UN Agenda: Framework, Metrics, Vocabulary"

The fourth step comprises three tasks (a, b c). As described in the Introduction, the UN Agenda and SDGs can act as a sensemaking frameworks to map goals and actions in relation to global sustainability. The core of SDGs is their role as metrics, that is, functioning as a tool for measuring sustainable development. However, in almost all cases, SDG targets and indicators need to be redesigned to serve a more specific context, global or local. As SDGs are a global set of goals and the UN Agenda is a globally recognised agenda, the tool also functions as a vocabulary to communicate different goals locally and worldwide.

In task 4a, the UN Agenda 2030 and SDG roles (framework, metrics, vocabulary) are set in order of relevance to the chosen phenomenon. In addition, key words relating to the phenomenon can be added to each theme: framework, metrics, and vocabulary.

Time allocated for the task: approximately 5–15 min.

### Step 4b "Targets and Indicators"

This is one of the two main tasks in the process (in addition to 3a). It is close to a Hegelian dialectic approach, where synthesis is created via thesis and antithesis (Bisong and Oti 2021). The objectives are, first, to understand what the main definable goals of the selected phenomenon are, and second, to establish which of the SDG targets and indicators describe them. A synthesis of these two is then created. When successful, new indicators that connect the goals of the phenomenon and the SDGs have been created. These can be used both to measure and communicate the context-driven choices and local goals. Additionally, they have a direct link to the original SDGs. This acquired capacity and new competence are the primary rationale for the SDG sensemaking process.

Time allocated for the task: approximately 45–60 min.

### Step 4c "Handprint effect"

This task is one of the most recent additions to SST, and focuses on the Handprint effect. Handprint is defined as something that catalyses positive change or development of a given phenomenon. Participants are asked to define what, in their project or topic, is scalable and catalyses positive change.

Time allocated for the task: approximately 5–15 min.

### Step 5 "Strategic, tactical and operative levels"

In Step 5 (one task), the strategic, tactical and operative elements of the analysed phenomenon are identified and explicated. The goal is to make explicit which areas of the phenomenon have a direct link to city-level strategy, which are linked to the project or action-plan level, and which are operative and part of day-to-day practice.

1. Strategic: Direct link to city strategy
2. Tactical: Project plan/action plan
3. Operative: Part of daily practice

Time allocated for the task: approximately 5–10 min.

Step 6 "Network and role of the UN Agenda"

Step 6 (one task) explicates social networks, different roles and an understanding of your own role in different dimensions of the UN Agenda (framework, metrics, vocabulary). Beyond that, all relevant stakeholders are added to the matrix (see Figure 4). Their roles and functions are classified as:

(1) Observe: Persons who know the content/project but do not participate in it;
(2) Participate: Persons who are actively engaged in the work/project but not in charge of it; and
(3) Responsible for coordination: Persons who are in charge of the work/project.

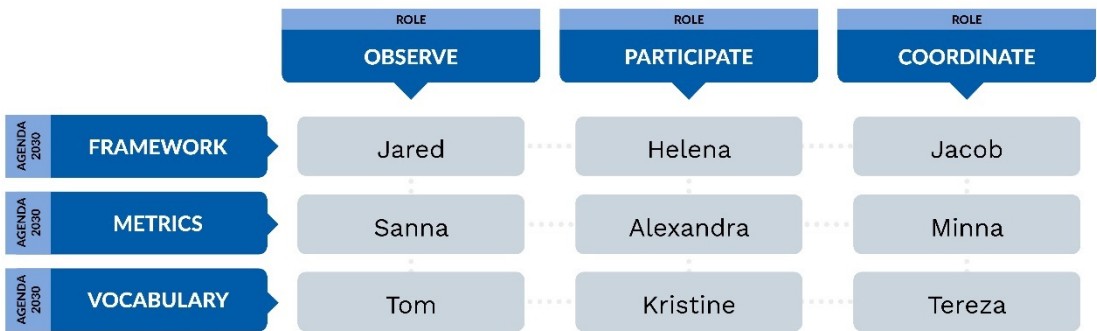

**Figure 4.** Step 6: identifying project network in relation to UN Agenda 2030 roles (Step 4a).

The idea of this task is to generate discussions and compare results, which will add an understanding of each person's roles. Pilot workshop results have shown that there are differences between participants in how the matrix is filled. This creates fruitful discussions and adds to team cohesion (please see Figure 4).

Time allocated for the task: approximately 10–15 min.

Step 7 "Threats and opportunities, dialogue and concrete next steps"

One of the fundamental goals of the SDG Sensemaking tool and analysis process is to promote the dialogue of and between different stakeholders, in relation to an observed and investigated phenomenon or project and the UN Agenda 2030 and SDGs (please see Figure 5). Due to its holistic and complex nature, a key challenge in sustainable development is more about understanding what needs to be solved rather than just solving it. Solutions tend to be simple and linear only after they have been properly observed and understood. In the last task of the workshop (Step 7), a dialogue is activated between participants in order to make sense of what concrete steps can and should be taken to implement results, what advances the process, and what hinders it. The time allocated to this task depends on the participants, and can vary from 15 min to 1 h.

1. What issues make achieving the SDGs difficult?
2. What issues help in achieving the SDGs?
3. What concrete steps am I going to take to achieve the SDGs in my work?

Time allocated for the task: based on discussion.

After defining the main and supporting SDGs, the sustainability dimensions and operative environment are Analysed. In Step 4 (b), new Localised indicators are created, while steps 5 and 6 are about identifying and defining dimensions of strategy, roles and the network.

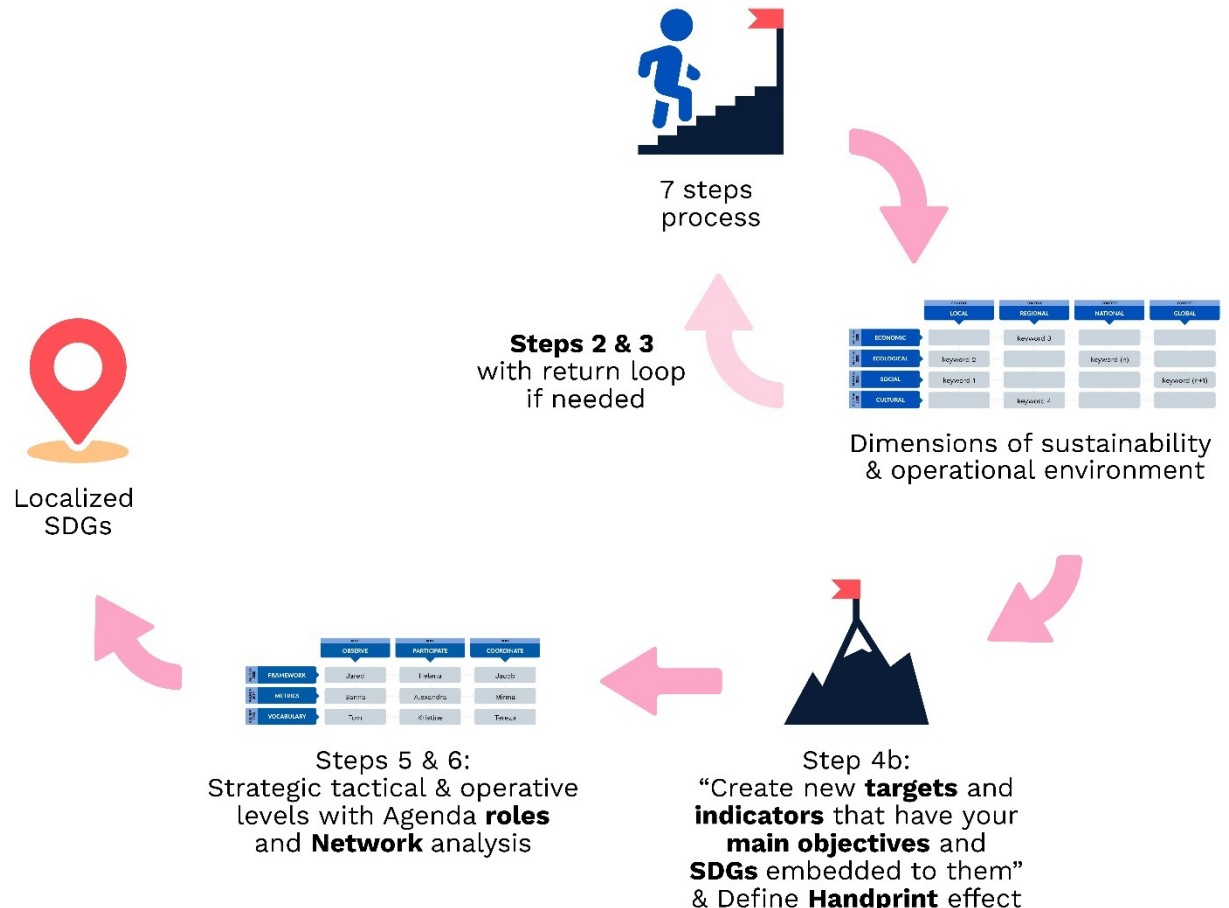

**Figure 5.** The main steps of the SST process.

### 4.3. Learnings from Workshops and Co-Developer Experiences

The SST is still very much a work-in-progress. Feedback from expert sessions (17), full pilot workshops (4), semi-structured interviews with SST pilot participants (two interviews, altogether six persons), and from partner stakeholders and institutions has shaped SST considerably.

Based on these experiences, completing the SST process increased participants' understanding of the complex nature of sustainable development and SDGs. Similarly, it increased the individual participant's comprehension of the SDG targets and indicators, and their interconnected nature. Completing all seven steps creates a scaffold that makes it easier to map the defined objectives in relation to holistic sustainability and find solutions on how to proceed with planning and implementation. The SST process has also facilitated the creation of indicators that connect local objectives with global goals. New localised indicators results can be used to communicate and advance selected solution paths. Additionally, team members found that after the SST workshop they had a better understanding of how the other members of their team make sense of their roles in relation to the analysed phenomenon. This may help future cooperation within the team and improve communication with other teams or stakeholders.

However, the SST was found to carry a high cognitive burden. This means there might be a need to develop the tool to be either more case-dependent or streamlined and focus on key steps instead of the whole process. One possible solution could be to develop a more detailed tool along with a lighter version or to divide the workshops into shorter sessions. These could then be used for different target groups as appropriate.

Stakeholder understanding of the UN Agenda 2030 varies substantially, which raises the question of how much background information is needed prior to the workshop. This should be considered when organising SST workshops for non-experts. Although inter-

action amongst the participating team members was enhanced during the process, it is important to further develop ways in which to increase intra- and inter-team communication. Finally, the current MS Excel format for documentation was found to be non-intuitive and should be further developed and better designed.

## 5. Conclusions

The world is facing a myriad of urgent sustainability challenges that need to be mitigated and resolved. In addition to climate change and biodiversity loss, we are living through a time of global pandemics, refugee crises, political disruption, more extreme weather, and other ecological, economic and social challenges, arising at an accelerating pace (IPCC 2021). Cities can and should play a fundamental role in meeting and overcoming these challenges (Bettencourt 2021).

That said, achieving sustainability is as much, if not more, about motivation, intention and participation than purely technical solutions. To understand how to overcome complex challenges, we must first make sense of what the challenge is, and how it can be framed collectively across scales, especially in each local context (Lai 2020). The overarching research question in this practitioner article, "How to utilise UN Agenda 2030 and SDGs that are a multilateral, global, national and state level set of goals, within an urban context, i.e., in a city?" was approached from a pragmatic perspective, codifying a processual strategy towards an answer. More precise questions were formulated as the procedure was implemented in group discussions and workshops, and as the analysis progressed.

Addressing the question "What does it mean for a city to be committed to SDGs and their measurement?", we found that in order for SDGs to function as a framework, metrics, and a vocabulary, a city has to commit to UN Agenda 2030 at all levels, including the strategic, tactical, and operative. Only when there is top management engagement in the organization can the results of the SST process become fully useful in a local, regional, national, and international context. Our SST has, to date, been employed within city organizations. Future work and research will focus on how to engage municipal residents, businesses and other relevant stakeholders. This work will start in the immediate future, and the hypothesis is that communication and opportunities for stakeholder agency will play a significant role.

Next, addressing the question "How can a city use SDGs to make sense of, measure, monitor, and communicate both its present and future plans through SDGs aimed at nation states and the global level?", we found the SDG framework to be a good fit for sensemaking, measuring, monitoring, and communicating present and future city plans. However, to function well, it requires a systematic localization process that connects the more abstract international SDG targets and indicators to more specific local measures. Although not perfect or all-encompassing, UN Agenda 2030 and the SDGs answer the above question better than any other extant framework, standard or set of metrics.

The SST can be used as a framework to map needs, measure and steer action, and for communication within and outside a city. It can also be employed to create new indicators linking local objectives to international SDG targets and indicators. As noted in the previous section, the future focus will be on, first, how to engage citizens and other stakeholders in the process, and second, how to communicate results to citizens in a way that promotes dialogue and learning.

Finally, in addressing the question "How does the tool manage to first make sense of and then reduce complexity, and thereby make the wicked problems of (sustainable) development simpler and more actionable?", we found that the SST process increases the general understanding of what the global UN Agenda 2030 means locally in a chosen context. It also facilitates the creation of shared understanding and interaction among participants on what UN Agenda 2030 means for a particular project, and how to utilise it in practice. This applies to all roles of the UN Agenda, that is, setting a framework, providing metrics, and creating a common vocabulary.

Participants of the SST sessions gain greater shared insights on how to achieve common goals through interaction. The processes clarified different dimensions of sustainable development (economic, social, ecological, cultural) in different contexts, and at local, regional, national, and international levels. As a result, we observed that developing understanding and knowledge, especially on SDG targets and indicators, and how other participating team members make sense of them, was increased. For core project teams, time invested in the SST workshop in relation to outcomes was found feasible and viable. For pilot participants, there was no need to pre-read UN Agenda 2030 (preamble, declaration, SDGs). However, further development on delivering the pre-workshop information is needed when (simplified) workshops are organised for citizens or municipal residents, or other participants not well-acquainted with UN Agenda 2030.

The SDG Sensemaking Tool (SST) proved to be a useful process in all our pilot cases. In the City of Espoo (the context wherein the SST has been developed), the SDGs have influenced the analysed phenomena and led to the creation of localised indicators (Task 4b). Workshops have shown that, in task 4b, working through the targets and indicators of the main SDG is demanding enough, while, ideally, this task should cover the supporting SDGs as well. Solving this overload problem requires further development of the SST.

The rationale for developing the SST was that although cities face similar challenges on a larger scale, the paths to solutions are case-dependent and vary. SST acts as a phenomenon-based process, which takes into account city specific needs, resources and culture. It aims for societal improvement at local and other levels. Policymakers, city officials, non-governmental organisations, and even companies can replicate the process in other cities, regions or different contexts and produce outcomes, which serve their specific development needs. The workshop results are at the same time transparent and comparable as they are induced using the same process; SST.

The preliminary results are promising, but a lot of work remains to be carried out. Future development will comprise two parallel paths. First, the further development of SST in different cities and, second, developing the use of SST to increase citizen engagement (in terms of both participation and feeling of belonging). In the latter track, one research theme with strategic relevance to SDG work is how Storytelling could be used as a way to communicate results and understanding created in the SST process (Karanian et al. 2019).

In addition to these steps in SST development, the six largest cities in Finland have committed to using the tool and the Ministry of Finance in Finland is funding its development until the end of 2023. In November 2021, the Ministry of the Environment announced that the City of Espoo will receive a two-year funding input for building a concept on how to engage citizens in local SDG work. This will enable future development of the SST, especially in research and action in track two.

The SST was also presented at the UN High-Level Political Forum in July 2021, where it generated interest from other nations. The SST is open access, open source and follows the Linux philosophy. Similar to the UN Agenda 2030 and SDGs, it belongs to everyone. In this way, we hope that the SST contributes to collaborative processes and to the improvement of methods towards localizing and accelerating global sustainable development that is meaningful to people in diverse environments, in a context appropriate way.

**Author Contributions:** Conceptualization, V.T. and B.K.; methodology, V.T. and M.J.; validation, V.T., M.J. and L.B.; formal analysis, V.T.; investigation, V.T. and M.J.; resources, L.B.; data curation, V.T. and M.J.; writing—original draft preparation, V.T. and M.J.; writing—review and editing, V.T., M.J., B.K., L.B.; visualization, V.T.; supervision, L.B.; project administration, V.T. and M.J.; funding acquisition, L.B. All authors have read and agreed to the published version of the manuscript.

**Funding:** Publication of this article was funded by the University of Chicago, USA.

**Institutional Review Board Statement:** The study did not require ethical approval.

**Informed Consent Statement:** Informed consent was obtained from all subjects involved in the study.

**Data Availability Statement:** All workshop and interview data can be found from corresponding author: ville.taajamaa@espoo.fi.

**Acknowledgments:** We wish to thank Minna Konttinen and Minna Ögland (City of Espoo) for their excellent work on Figures 1–5. Sincere thanks to Alexander Frost for his help with English language.

**Conflicts of Interest:** There are no conflict of interest.

## Note

1  espoo.fi/en (accessed on 10 February 2022).

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
