# Peer review of "Seven Steps to Strategic SDG Sensemaking for Cities"

_admsci, doi:10.3390/admsci12010033_

Round 1

Reviewer 1 Report

Review to admsci-1494229

Seven Steps to Strategic SDG Sensemaking for Cities

Content

The paper presents an analysis process that has tangible strategic, tactical and operative context-driven outcomes that can be utilised to further the achievement of SDGs in cities for solving the social, ecological and urban impacts.

General Comments

An exciting and important contribution to the planning community by the title and the prospect of solving one of the most fundamental problems of "sustainable urban planning", the "HOW to deal with the SDGs".Arouses direct interest through the title, although unfortunately many questions remain unanswered

Starting with the formal:
Classically divided into an introduction, research question, theory and method section, results and conclusion. However, in principle, I would merge the theory part, i.e. chapters 3, 4 and 5 as one chapter 3.1, 3.2 and 3.3 - > the sections become too important, and on the whole, provide too little content for a "separate" chapter.

Also, For the whole article, the question arises anyway whether it would not make more sense to formulate the questions posed in 7.1.1 as "headliners" in chapter 2, then this would also be more comprehensible. because in reality, there is a lot of "introduction" in chapter 2 and only ONE question, which is very general and does not pose the strategic questions from 7.1.1, which, in my opinion, form the core of the article.

In my opinion, tables 1 and 2 (screenshot from Excel?) do not help to understand the tool, as they only show the time required for the individual steps, but do not explain anything. Please place references in the text BEFORE the punctuation mark.

calling chapter 6 a "methodology chapter" also implies very strong importance that is not given here. It describes the process, the workflow, which components were carried out, but how the whole thing is transferred to the SST is not shown here, and in my opinion, this is where the method would start, HOW was it implemented? As a compromise: don't call it a "method", but a process and the story behind it, then it is more honest.

Everything else about the basics of this contribution is in the section of the Overall recommendation.

Comments in Detail

l.35: really “pan”-disciplinary, or, especially in this practitioner section better “inter”-disciplinary or a holistic approach? Or both.

124: If "wicked" problems are mentioned here, I would like to see the "forefather" quoted once, even in a practitioner's report, especially since the Finnish sources are not helpful here (perhaps translate the title in the bibliography) and these certainly also refer to Rittel.

125 ff: Especially in the context of the "local" urban planning application domain, what about the "new urban agenda", where Habitat III and SDGs are also strongly interlinked, if only from the evolution of the frameworks. please also include this.

Overall Recommendations

Accepting after major revisions.

I think the biggest problem with this contribution is that the authors don't know WHAT kind of contributions they want to write here.

For a practical article "too scientific" at the beginning with too little information about the whole SST process and above all the effects and how it is applied e.g. in a concrete example. Too imprecise for a (shortened) scientific article.

Overall, the common thread within the article is missing, which takes up a very important topic, creates awareness through the described steps of the SST, but repeatedly fails to address the important information for the reader through the sole descriptive.

I would find it much more exciting, and I am sure that the authors have applied the knowledge to see "how" successful the tool is with the participants, i.e. how the goals slowly solidify as an important planning goal, also how the participants are guided (how do I identify the goals, how the operational etc.).

How much time does it take for implementation with the participants? How does the city leadership see this? Could any missing integration of goals be adjusted afterwards? AND above all: how does the tool manage to reduce complexity, and thereby also make the wicked problems seem less (?) wicked at least in part?

I would like to see the authors take this courageous step of reorganising the paper, as I see a very valuable contribution to planning practice between the lines here, but lose it more and more when reading it due to the structure of the paper and the somewhat "hectic" and unfortunately inconsistent execution.

Literature

Rittel, H. W. J., & Webber, M. M. (1973). Dilemmas in a general theory of planning. Policy Sciences, 4(2), 155–169. https://doi.org/10.1007/BF01405730

Rith, C., & Dubberly, H. (2007). Why Horst W. J. Rittel Matters. Design Issues, 23(1), 72–91. https://doi.org/10.1162/desi.2007.23.1.72

Author Response

Dear Sir/Madame,

We are very grateful for your review, comments and suggestion to our paper. The feedback was very valuable and helped us a lot. 

The whole practitioner article is now re-structured and for most part re-written. Altogether close to 1100 changes has been made. Also several new graphs were designed and implemented. 

We thank you for your valuable feedback. Please find a revised version of the article attached to this email. 

Sincerely and on behalf of our team,

corresponding author

point-by-point answers to review-comments can be found below

Classically divided into an introduction, research question, theory and method section, results and conclusion. However, in principle, I would merge the theory part, i.e. chapters 3, 4 and 5 as one chapter 3.1, 3.2 and 3.3 - > the sections become too important, and on the whole, provide too little content for a "separate" chapter.

The whole structure of the article is re-organized. Please refer to the revised version

For the whole article, the question arises anyway whether it would not make more sense to formulate the questions posed in 7.1.1 as "headliners" in chapter 2, then this would also be more comprehensible. because in reality, there is a lot of "introduction" in chapter 2 and only ONE question, which is very general and does not pose the strategic questions from 7.1.1, which, in my opinion, form the core of the article.

The whole structure of the article is re-organized and ´Research questions´ part is re-written to better describe the actual process.  Please refer to the revised version

In my opinion, tables 1 and 2 (screenshot from Excel?) do not help to understand the tool, as they only show the time required for the individual steps, but do not explain anything.

All tables and figures are re-designed

Please place references in the text BEFORE the punctuation mark.

references are before punctuation

calling chapter 6 a "methodology chapter" also implies very strong importance that is not given here. It describes the process, the workflow, which components were carried out, but how the whole thing is transferred to the SST is not shown here, and in my opinion, this is where the method would start, HOW was it implemented? As a compromise: don't call it a "method", but a process and the story behind it, then it is more honest.

methodology part is re-written to describes the actual content better.

l.35: really “pan”-disciplinary, or, especially in this practitioner section better “inter”-disciplinary or a holistic approach? Or both.

the word holistic is used when ever suitable

124: If "wicked" problems are mentioned here, I would like to see the "forefather" quoted once, even in a practitioner's report, especially since the Finnish sources are not helpful here (perhaps translate the title in the bibliography) and these certainly also refer to Rittel.

Required references are added and Finnish references are translated to English

125 ff: Especially in the context of the "local" urban planning application domain, what about the "new urban agenda", where Habitat III and SDGs are also strongly interlinked, if only from the evolution of the frameworks. please also include this.

New urban agenda Habitat III is added

I think the biggest problem with this contribution is that the authors don't know WHAT kind of contributions they want to write here.

For a practical article "too scientific" at the beginning with too little information about the whole SST process and above all the effects and how it is applied e.g. in a concrete example. Too imprecise for a (shortened) scientific article.

Overall, the common thread within the article is missing, which takes up a very important topic, creates awareness through the described steps of the SST, but repeatedly fails to address the important information for the reader through the sole descriptive.

I would find it much more exciting, and I am sure that the authors have applied the knowledge to see "how" successful the tool is with the participants, i.e. how the goals slowly solidify as an important planning goal, also how the participants are guided (how do I identify the goals, how the operational etc.).

Whole results part is re-written together with the whole content and structure. SST part is re-written

“how” successful part is dealt with by defining better and in more detail what was done and also in future research part

How much time does it take for implementation with the participants? How does the city leadership see this? Could any missing integration of goals be adjusted afterwards? AND above all: how does the tool manage to reduce complexity, and thereby also make the wicked problems seem less (?) wicked at least in part?

above points will be dealt with more detail SST part

Reviewer 2 Report

  1. I’ve read the paper thoroughly. It focuses on the important issue of urban development but there are several major issues that have to be addressed in the review.
  2. The paper tries to explain a „need to make sense of UN Agenda 2030 and Sustainable Development Goals (SDGs) at a city level and in an urban context, and secondly on a need to explain how to utilise them in strategic, tactical and operative urban development”. Such ambitious goals need a precise research design to be fulfilled. Therefore a short subchapter on the research process designed for this paper should be presented in the introductory part.
  3. Author(s) precisely identified a knowledge gap in “how to localise the Sustainable Development Goals to an urban context”. This is a strong point of the paper.
  4. [Rows 38-40] Why do Author(s) select exactly such sources to discuss the topic of “the finding the right questions about sustainable development is at least as important as finding answers to it?” There is a huge literature on this subject, therefore it is important to find out, why these publications were selected.
  5. Although it is a very good practice that Author(s) underlined research questions by the separate subchapter (nr 2), the title is not very appropriate as the subchapter contains in fact only one research question.
  6. Moreover, the research question (“How to utilise Agenda 2030 and SDGs that are a multilateral, global, national and state-level set of goals, within an urban context, i.e. in a city?”) is rather general and lacks further detailing. In my opinion, such a general research question results in several disadvantages of the paper I describe below.
  7. A general remark is that Chapter 3 is not enough substantiated by the relevant literature. Although the division of levels of complexity facing cities into three levels is very good and well structured (a very strong point of this subchapter!), the description and explanation of these levels lack deepening based on the relevant and more extensive literature review. Some kind of inspiration for the improvement of this part could be found in the Editorial to “Journal of Urban Management” 2020 nr 9 titled “On the solvability of urban complexity” by Shih-Kung Lai.
  8. [Rows 104-105] In my opinion underlining the difference between “ most European cities” and “the megalopolises in Asia” is unjustified from the scientific point of view. Even in Europe cities vary according to size, area, functions, political and economic importance as well as legal and managerial forms. Also, Asian cities represent a variety of urban forms with different sizes and populations. Megalopolis is only one of them.
  9. The SDG Sensemaking Tool (SST) should be prepared in a better graphic form. The Tool description lacks an objective presentation of its advantages and disadvantages. Moreover, a direct link to the Tool shall be provided.
  10. [rows 403-422] It is difficult to accept the fact of the complex assessment of the Tool that was based on only 2 interviews without any written prove (i.e. research report, evaluation report…). From the scientific point of view, it is a very serious shortcoming of the paper, badly influencing its methodological aspect.
  11. Also, “Key findings from co-developers at University La Salle, Chile” [rows 434-452] are not supported by the relevant source.
  12. In Conclusion, the Author(s) stated that “Process was developed in a Nordic city, namely the city of Espoo in Finland” [rows 482-483]. In my opinion, it would be valuable to discuss the exact case study of the Espoo instead of presenting the process of the tool development supported by unclear (from the scientific point of view) sources.
  13. Also, in the same subchapter. Author(s) stated that “The SDG Sensemaking Tool (SST) has been proven to be a useful process in all piloted cases” [rows 488-489]. Do these “cases” include Espoo, Chilean cities or others? If the Author(s) mention concrete cases, they should present them clearly. The lack of a precise methodological approach is another important shortcoming of this paper. Although Author(s) claimed that “This paper will present preliminary findings from this development process and pilots”, I could not easily find the “pilots” for the implementation.
  14. Continuing this topic, the Author(s) claim that “this paper presents an analysis process that has tangible strategic, tactical and operative context-driven outcomes” [rows 59-60]. What are they?

Technical remarks:

  1. [row 199-200] “The SDG Sensemaking Tool (SST) was developed to answer a specific need in a growing Northern European city, namely the city of Espoo (espoo.fi)”. Providing a general link to the official website of Turku city is, in my opinion, not enough. A direct link to the Tools shall be provided.
  2. The list of references does not present all publications quoted in the manuscript (i.e. Jacobs 1961, Dijkstra et al. 2018).
  3. The style of the text in rows 262-263 is partly not appropriate for the scientific publication (“They can also be used as a vocabulary of sustainable development, a ´lingua franca´ if you will”).

Suggested major changes:

  1. I recommend redeveloping the paper in the following way:
    • Subchapter 2 should change its title (i.e. Methodological approach). The whole research design being used in this paper should be clearly presented. If there are concrete cases – they should be mentioned (i.e. city of Espoo if needed). Then, the final part of this subchapter should be summarized with the research question.
    • The research question itself should be more precise.
    • When Author(s) use case study analysis, it would be strongly recommended to describe particular cities and the results that have been achieved due to the Tool development.
    • As Chapter 3 is well structured into three levels it should be improved with an extensive literature review.
    • The description of the process of the Tool development should be shortened. Author(s) should think about improvement of the graphic part of the paper as it lacks quality.
    • Proper citation should be included wherever appropriate - written evidence of the results is a fundament of the research.
    • The conclusion part should be rewritten as part of the paper would change.

Author Response

Dear Sir/Madame,

We are very grateful for your review, comments and suggestion to our paper. The feedback was very valuable and helped us a lot. 

The whole practitioner article is now re-structured and for most part re-written. Altogether close to 1100 changes has been made. Also several new graphs were designed and implemented. 

We thank you for your valuable feedback. Please find a revised version of the article attached to this email. 

Sincerely and on behalf of our team,

corresponding author

point-by-point answers to review-comments can be found below

[Rows 38-40] Why do Author(s) select exactly such sources to discuss the topic of “the finding the right questions about sustainable development is at least as important as finding answers to it?” There is a huge literature on this subject, therefore it is important to find out, why these publications were selected.

new sources are added

Although it is a very good practice that Author(s) underlined research questions by the separate subchapter (nr 2), the title is not very appropriate as the subchapter contains in fact only one research question.

The whole structure of the article is re-organized. Please refer to the revised version

Moreover, the research question (“How to utilise Agenda 2030 and SDGs that are a multilateral, global, national and state-level set of goals, within an urban context, i.e. in a city?”) is rather general and lacks further detailing. In my opinion, such a general research question results in several disadvantages of the paper I describe below.

Research questions are re-written

A general remark is that Chapter 3 is not enough substantiated by the relevant literature. Although the division of levels of complexity facing cities into three levels is very good and well structured (a very strong point of this subchapter!), the description and explanation of these levels lack deepening based on the relevant and more extensive literature review. Some kind of inspiration for the improvement of this part could be found in the Editorial to “Journal of Urban Management” 2020 nr 9 titled “On the solvability of urban complexity” by Shih-Kung Lai.

more literature review was done

[Rows 104-105] In my opinion underlining the difference between “ most European cities” and “the megalopolises in Asia” is unjustified from the scientific point of view. Even in Europe cities vary according to size, area, functions, political and economic importance as well as legal and managerial forms. Also, Asian cities represent a variety of urban forms with different sizes and populations. Megalopolis is only one of them.

this sentence was restructured

The SDG Sensemaking Tool (SST) should be prepared in a better graphic form. The Tool description lacks an objective presentation of its advantages and disadvantages. Moreover, a direct link to the Tool shall be provided.

example graphics were added

[rows 403-422] It is difficult to accept the fact of the complex assessment of the Tool that was based on only 2 interviews without any written prove (i.e. research report, evaluation report…). From the scientific point of view, it is a very serious shortcoming of the paper, badly influencing its methodological aspect.

a more descriptive reference and more data and graphs on how feedback was gathered will be added

Also, “Key findings from co-developers at University La Salle, Chile” [rows 434-452] are not supported by the relevant source.

In Conclusion, the Author(s) stated that “Process was developed in a Nordic city, namely the city of Espoo in Finland” [rows 482-483]. In my opinion, it would be valuable to discuss the exact case study of the Espoo instead of presenting the process of the tool development supported by unclear (from the scientific point of view) sources.

This part was re-written. Please refer to the article

Also, in the same subchapter. Author(s) stated that “The SDG Sensemaking Tool (SST) has been proven to be a useful process in all piloted cases” [rows 488-489]. Do these “cases” include Espoo, Chilean cities or others? If the Author(s) mention concrete cases, they should present them clearly. The lack of a precise methodological approach is another important shortcoming of this paper. Although Author(s) claimed that “This paper will present preliminary findings from this development process and pilots”, I could not easily find the “pilots” for the implementation.

more text on how piloted cases and other feedback was gathered was added

Continuing this topic, the Author(s) claim that “this paper presents an analysis process that has tangible strategic, tactical and operative context-driven outcomes” [rows 59-60]. What are they?

explanation on strategic, tactical and operative context-driven outcomes was  be elaborated on and explained more thoroughly

[row 199-200] “The SDG Sensemaking Tool (SST) was developed to answer a specific need in a growing Northern European city, namely the city of Espoo (espoo.fi)”. Providing a general link to the official website of Turku city is, in my opinion, not enough. A direct link to the Tools shall be provided.

The style of the text in rows 262-263 is partly not appropriate for the scientific publication (“They can also be used as a vocabulary of sustainable development, a ´lingua franca´ if you will”).

This part was re-written

The list of references does not present all publications quoted in the manuscript (i.e. Jacobs 1961, Dijkstra et al. 2018).

references were checked

Subchapter 2 should change its title (i.e. Methodological approach). The whole research design being used in this paper should be clearly presented. If there are concrete cases – they should be mentioned (i.e. city of Espoo if needed). Then, the final part of this subchapter should be summarized with the research question.

more feedback, data and graphs were added.

The conclusion part should be rewritten as part of the paper would change.

conclusion part was re-written

Proper citation should be included wherever appropriate - written evidence of the results is a fundament of the research.

citation was checked

The description of the process of the Tool development should be shortened. Author(s) should think about improvement of the graphic part of the paper as it lacks quality.

all graphs were redesigned

As Chapter 3 is well structured into three levels it should be improved with an extensive literature review.

more literature review were added

When Author(s) use case study analysis, it would be strongly recommended to describe particular cities and the results that have been achieved due to the Tool development.

this part was re-written

The research question itself should be more precise.

research questions were re-designed

Reviewer 3 Report

At first I would like to thank the authors for the opportunity of reading and commenting their work. The topic in debate is very interesting and actual. Also, presenting the contribution from a practitioner perspective is very enriching and different from research papers.

That being said, I believe that the present form of the paper is not ready for publication, but some improvements need to be made to grant the document a new dimension .

Below, some comments will be raised to contribute towards the improvement of this version: 

  1. concerning the writing style, the authors should avoid the use of unnecessary adjectives. The document needs to be revised as some paragraphs are hard to follow. Also some parts of the document could be more concise as they encompass too many noramtive statements, which is undesirable in scientific papers.
  2. the formatting looks very sloppy, mainly in what relates to the tables. Also, the author mentions "picture" and I believe that it shoulf be figure/image. Needs deep improvement. The same applies for the bullet points along the different sections - this style is hardly found in such an extensive use in scientific papers.
  3. The contextualization of the extant literature is too short to provide a theoretical foundation for the concept in test. 
  4. Also, there is a need for further clarification in terms of which pillar of sustainability is about to be explored, as there is common acceptance of the singularities of each dimension. 
  5. A very recent paper from MDPI, can be of use to discuss the dimensions and the simultaneity in their achievement: Costa, J.; Cancela, D.; Reis, J. Neverland or Tomorrowland? Addressing (In)compatibility among the SDG Pillars in Europe. Int. J. Environ. Res. Public Health 202118, 11858.
  6. Along the paper too many questions are posed, and not a direct reseponse is provided. Please reformulate.
  7. The purpose of the different "tools" in the study is not very clearly explained. No direct connection to the literature is made to grant the respect to the methods.
  8. Conceptualization takers too many points/sections in the paper. All these aspects should be re-organized and put under the umbrell of the literature review.
  9. The results section normally is the third/fourth section of such work.
  10. The results lack systematization. There is a need for reformulation.
  11. The presentation of the steps needs further debate and clarification about the methodological choices in data collection.
  12. Then the authors jump to the conclusions, without any previous expectation about what was going to happen next. Please reconsider changing.
  13. How can the reader check the value of several statements if no referencing is made to the grounded theory.
  14. Conclusions need to provide a broader picture of the results of this sort of experiment.
  15. There is also a need to reinforce the findings of the paper. And, a clarification about the implications of these initiatives as well as the projects.

Best of luck with your research!

Hope that you find my comments usefull to the improvement of your document.

Author Response

Dear Sir/Madame,

We are very grateful for your review, comments and suggestion to our paper. The feedback was very valuable and helped us a lot. 

The whole practitioner article is now re-structured and for most part re-written. Altogether close to 1100 changes has been made. Also several new graphs were designed and implemented. 

We thank you for your valuable feedback. Please find a revised version of the article attached to this email.  

Sincerely and on behalf of our team,

corresponding author

short answers to review comments point-by-point below:

  1. concerning the writing style, the authors should avoid the use of unnecessary adjectives. The document needs to be revised as some paragraphs are hard to follow. Also some parts of the document could be more concise as they encompass too many noramtive statements, which is undesirable in scientific papers.

writing style, structure and content was revised. please refer to the revised version.

  1. the formatting looks very sloppy, mainly in what relates to the tables. Also, the author mentions "picture" and I believe that it shoulf be figure/image. Needs deep improvement. The same applies for the bullet points along the different sections - this style is hardly found in such an extensive use in scientific papers.

formatting was re-structured, all figures, pictures and graphs were re-designed

  1. The contextualization of the extant literature is too short to provide a theoretical foundation for the concept in test.

theoretical part of the paper was re-designed and re-written

  1. Also, there is a need for further clarification in terms of which pillar of sustainability is about to be explored, as there is common acceptance of the singularities of each dimension. 

sustainability dimensions were elaborated on.

  1. A very recent paper from MDPI, can be of use to discuss the dimensions and the simultaneity in their achievement: Costa, J.; Cancela, D.; Reis, J. Neverland or Tomorrowland? Addressing (In)compatibility among the SDG Pillars in Europe. Int. J. Environ. Res. Public Health 202118, 11858.

above-mentioned and other relevant papers were checked and read.

  1. Along the paper too many questions are posed, and not a direct reseponse is provided. Please reformulate.

questions and other content were reformulated

  1. The purpose of the different "tools" in the study is not very clearly explained. No direct connection to the literature is made to grant the respect to the methods.

methods part was re-designed and re-written

  1. Conceptualization takers too many points/sections in the paper. All these aspects should be re-organized and put under the umbrell of the literature review.

literature review part was re-written

  1. The results section normally is the third/fourth section of such work.

whole results section was re-organised to learnings and re-written

  1. The results lack systematization. There is a need for reformulation

whole results section was re-organised to learnings and re-written

  1. The presentation of the steps needs further debate and clarification about the methodological choices in data collection.

all steps were re-written

  1. Then the authors jump to the conclusions, without any previous expectation about what was going to happen next. Please reconsider changing.

This part was re-structured and re-written

  1. How can the reader check the value of several statements if no referencing is made to the grounded theory.

theoretical bases and epistemology of research was re-written

  1. Conclusions need to provide a broader picture of the results of this sort of experiment.

Conclusions part was re-written

  1. There is also a need to reinforce the findings of the paper. And, a clarification about the implications of these initiatives as well as the projects.

results and learnings part was re-designed and re-written

Round 2

Reviewer 1 Report

Thank you very much for the extensive work on the revisions.

The article is now much more structured and "sells" exactly what is "there", what the project has achieved so far and gives an outlook on what it could achieve in the future.

I think the mix of academic writing and practical descriptions of the process and work can now help other practitioners to better understand SDG integration (with the help of a tool, maybe in the future also with this tool), to learn from the "Espoo Story" and possibly to better integrate it in their city.

Thank you for the intensive work, it was worth it.

Author Response

Thank you for your very valuable insight and comments.

Sincerely,

Author team 

Reviewer 2 Report

The manuscript has been rewritten to meet the majority of the Reviewer's remarks. I accept and appreciate the individual opinion of the Author.

Author Response

(The authors gave the same response as above.)

Reviewer 3 Report

Thanks for the response, however I felt very disappointed as the majority of my comments were not taken into consideration. I respect the author's point of view, however, my job is to comment accordingly. 

Please consider my comments and answer them properly. I have invested my time in your work and respected your points of vie. I expect the same. 

Author Response

Answers to REVIEW #3

  1. concerning the writing style, the authors should avoid the use of unnecessary adjectives. The document needs to be revised as some paragraphs are hard to follow. Also some parts of the document could be more concise as they encompass too many noramtive statements, which is undesirable in scientific papers.

Thank you for your comments. We have revised the writing style, structure and content. The article has also been checked by a native English speaker.

  1. the formatting looks very sloppy, mainly in what relates to the tables. Also, the author mentions "picture" and I believe that it shoulf be figure/image. Needs deep improvement. The same applies for the bullet points along the different sections - this style is hardly found in such an extensive use in scientific papers.

We have re-structured the formatting. Additionally, all figures, pictures and graphs were re-designed and re-drawn. We have reduced the amount of bullet points and clarified the flow of the article.

  1. The contextualization of the extant literature is too short to provide a theoretical foundation for the concept in test.

We revised, rewrote and redesigned the theoretical part of the paper. Considering the nature of the paper, we decided to keep it short.

  1. Also, there is a need for further clarification in terms of which pillar of sustainability is about to be explored, as there is common acceptance of the singularities of each dimension. 

SST is a tool that analyses all dimensions/pillars of sustainability: economic, ecological and social. In addition, strategic, tactical and operative dimensions are analysed. Also the network of people influencing the theme is analysed. In addition, geographical dimensions are analysed: local, regional, national and international. Future research will focus on citizen engagement through storytelling. This work will also have all pillars of sustainability embedded into them.

  1. A very recent paper from MDPI, can be of use to discuss the dimensions and the simultaneity in their achievement: Costa, J.; Cancela, D.; Reis, J. Neverland or Tomorrowland? Addressing (In)compatibility among the SDG Pillars in Europe. Int. J. Environ. Res. Public Health 202118, 11858.

The abovementioned article was helpful to this study and writing process. Also a reference of the article was added to section 2.2. Below are some excerpted points that we found especially interesting.

(p2) The proposed SDGs are often believed as too ambitious as they concentrate on environmental issues, as well as economic inclusion throughout inequality minimization, while relying upon economic development to foster this civilizational shift promoted in a sustainable ecosystem. The prior eight millennium goals were considered as focused and achievable; hence, these 17 SDGs are believed to be comprehensive and broad [19], becoming eventually unfeasible [20,21].

(conclusions) As a final remark, and based on the empirical results, one can state that there is an urgent need of a careful, rigorous and effective science-based analysis of the strategical tools implemented. Given the growing climate and environmental threats, there are indeed ways to harmonize material effluence with environmental preservation [18]. It is worth considering that addressing some SDGs may endanger others [9]. As a consequence, sustainable development is at risk of becoming a cliché, to whom everybody mentions but nobody seems to define, improve or implement [29,45]

(conclusion) Moreover, economic sustainability needs to be strongly rooted in innovation practices. Once more, developed nations need to consider knowledge sharing as a positive externality to their individual efforts, as the improvements in energy and production will benefit all. Notwithstanding, there is a need to develop policy action aiming to compensate these knowledge frameworks and the promotion of a solid globalized ecosystem.

  1. Along the paper too many questions are posed, and not a direct reseponse is provided. Please reformulate.

Questions part (and also other content were reformulated). Please see section Research Questions from page 5 for further review. The whole questions part is re-written to better suit what was researched.

  1. The purpose of the different "tools" in the study is not very clearly explained. No direct connection to the literature is made to grant the respect to the methods.

There is altogether new section (section 3) that explains the tool and learnings from product development, prototyping and piloting phases. The underlying science and literature are explained in previous section and partly in section 4 Conclusions. All in all, methods part was re-designed and re-written.

  1. Conceptualization takers too many points/sections in the paper. All these aspects should be re-organized and put under the umbrell of the literature review.

Main literature review is now in section 1 and section 2. The bulk of literature concerning this topic and background for SST is reviewed but kept short considering the nature of this paper (practitioner). All in all, literature part was re-designed and re-written.

  1. The results section normally is the third/fourth section of such work.

As this is a practitioner -based article with emphasis to the tool it presents, we decided to remove the whole Results section and structure it to learnings section as part of section 3, which explains the tool. All in all, this part was re-designed and re-written.

  1. The results lack systematization. There is a need for reformulation

Please refer to previous point (9). Whole Results section was re-organised to learnings and re-written. Also the content of Learnings was re-written

  1. The presentation of the steps needs further debate and clarification about the methodological choices in data collection.

All steps were re-written and data collection part was also re-written together with methodological choices and epistemological background. Excerpt from the article:

Experts from different domains of sustainable development within the city organisation gave feedback and comments on the early versions of SST in the beginning of 2021 and the first pilot workshops were held on 24 January 2021. Altogether, 17 expert sessions for testing the SST were held between 12 January and 1 March. Sessions lasted from 15 minutes (quick feedback reviews and comments) to three-hour workshops. In all the sessions except one, the participants were city officials. After each session the comments, critique and experiences were discussed and evaluated, and the draft tool was updated accordingly.”

  1. Then the authors jump to the conclusions, without any previous expectation about what was going to happen next. Please reconsider changing.

This part was re-structured and re-written: new order is first introduction, literature review, presentation of the tool coupled with learnings and finally conclusions.

  1. How can the reader check the value of several statements if no referencing is made to the grounded theory.

Theoretical bases and epistemology of research was re-written. Please refer to section 1 Introduction and Research design (page 3). Excerpt from the article:   “The epistemology of this practitioner article is founded on a pragmatic worldview (Brydon-Miller et al 2003): the starting point of this research is the problem of localizing the Sustainable Development Goals (SDGs) to an urban context. We selected research and analysis tools applicable to this specific problem and decided to start the process by creating a draft version of SDG Sensemaking Tool (SST). Following this step, we started a systematic testing process in expert sessions and workshops. The SST was modified according to the experiences and feedback generated in these workshops (please see Figure 1).”

  1. Conclusions need to provide a broader picture of the results of this sort of experiment.

Conclusions part was altogether re-written. Please refer to the new text starting from page 17 of the article.

  1. There is also a need to reinforce the findings of the paper. And, a clarification about the implications of these initiatives as well as the projects.

The whole results and learnings part was re-designed and re-written: Excerpt from the article below. We also tried to further clarify implications such as :

  • First, further development of SST in different cities. Second, developing use of SST to increase citizen engagement (participation and feeling of belonging)…

  • .. how could Storytelling be used as a way to communicate results and understanding created in the SST process?

excerpt: (page 17) Learnings from..:

Based on the experiences, completing the SST process has increased participants’ understanding of the complex nature of sustainable development and SDGs. Similarly, it has increased the participant’s comprehension of the SDG targets and indicators and their interconnected nature. Completing all Steps has created a scaffolding that makes it easier to map defined objectives in relation to holistic sustainability and to find solutions on how to proceed with planning and implementation. The SST process has also enabled creation of indicators, which connect local objectives with global goals. These results are seen as something that can be used to communicate selected solution paths forward. Additionally, team members also found that after the SST workshop they better understood how the other members of their team make sense of their roles in relation to the analyzed phenomenon. This may help future cooperation within the team and improve communication with other teams or stakeholders.

However, the SST was found to be very cognitively burdensome. This means, that there might be a need to develop the tool to be either more case-dependent or streamlined and focus on key Steps instead of the whole process. A possible solution could be to develop more detailed tool and a lighter version, which then can be used for different target groups accordingly.

Stakeholder understanding of UN Agenda 2030 varies, which raises the question of the amount of background information needed before the workshop. This should be considered when organising SST workshops to non-experts. Although interaction amongst the participating team members was enhanced during the process, it is important to further develop the ways to increase communication between team members and different teams. Finally, the current MS Excel format for documentation was found non-intuitive and should be further developed and better designed.

Round 3

Reviewer 3 Report

Many thanks to the authors for taking the time of revising and improving their article. I believe that the present version of the document is more organized and valuable than the former. 

As a general comment, in my opinion, adding the figures encompassing the steps improved the comprehension of the method employed and allows replicability. There should be a final effort to add a few paragraphs explaining the importance of their project to the achievement of the SDGs and its worth as a tool for societal improvement. How far can policymakers should replicate the process in other regions and what are the expectable outcomes. What are the lessons to be learned by other regions?

As a detailed comment, I would like to ask the authors to read the paper once more and correct the typos, shorten some sentences, and re-number the references that are incorrectly numbered - they go from 1 to 18 and then they start again on 1.

Best of luck with your research!

Author Response

 There should be a final effort to add a few paragraphs explaining the importance of their project to the achievement of the SDGs and its worth as a tool for societal improvement. How far can policymakers should replicate the process in other regions and what are the expectable outcomes. What are the lessons to be learned by other regions?

Thank you for your comments, they are much appreciated. Both Introduction and Conclusions are re-written so that reviewers comments have been taken into consideration.

As a detailed comment, I would like to ask the authors to read the paper once more and correct the typos, shorten some sentences, and re-number the references that are incorrectly numbered - they go from 1 to 18 and then they start again on 1.

Text has been read through and all typos found are corrected and long sentences shortened. also all ´&´ signs has been changed to “and”. Reference numbering has been corrected.